# Phosphorus Recovery from Sewage Sludge as Struvite

**Javier Cañas** [1,2], **Silvia Álvarez-Torrellas** [1,*], **Blanca Hermana** [2] and **Juan García** [1,*]

[1] Catalysis and Separation Processes Group (CyPS), Chemical Engineering and Materials Department, Complutense University of Madrid, Avda. Complutense s/n, 28040 Madrid, Spain; jcanas@ucm.es

[2] Ecolotum, Energía Recuperable, S.L., Calle del Cidro 3, 28044 Madrid, Spain; blanca.hermana@ecolotum.es

\* Correspondence: satorrellas@ucm.es (S.Á.-T.); jgarciar@ucm.es (J.G.)

**Abstract:** Environmental legislation on waste management coupled with the potential for nutrient recovery are key factors encouraging the use of advanced treatment technologies to manage biosolids waste. In this context, phosphorus recovery from sewage sludge treated by a wet oxidation (WO) process was carried out in this work. High organic matter (up to 85% in COD) and total solids content (up to 75%) removal values were achieved by the WO process at elevated temperature (up to 300 °C) and pressure (up to 200 bar) conditions. The liquid and solid fractions found in the oxidation process effluent contain high amounts of phosphorus that can be recovered. This research aims to maximize its recovery in both liquid and solid fractions. In the liquid effluent, phosphorus was recovered (up to 90 mg P/L) by chemical precipitation as struvite ($MgNH_4PO_4 \cdot 6\,H_2O$), a slow-release fertilizer. In this case, P recoveries greater than 95% were achieved. Additionally, the solid fraction, analyzed after filtration and drying (68 mg P/$g_{solid}$), was treated by acid leaching, obtaining up to 60% phosphorus recovery. All phosphorus extracted was in orthophosphate form.

**Keywords:** phosphorus; sewage sludge; struvite; wet oxidation

## 1. Introduction

Wet oxidation (WO) is a well-known process for treating liquid wastes with high organic matter contents. This process has shown high efficiency and cost-effective behavior compared to those technologies applied to wastes too diluted to be incinerated but too concentrated or poorly biodegradable to be treated by the usual biological or chemical treatments [1–3]. These wastes include mainly organic sewage sludge from municipal or industrial wastewater treatment plants (WWTPs) with a solids content of up to 5%, without dewatering and/or drying the sludge. The WO process has been studied under different operating conditions, mainly related to temperature (100–330 °C) and pressure (40–230 bar). In this sense, using pure oxygen and maintaining the liquid phase allows the breaking of the C-C bonds of a large amount of toxic and hazardous organic compounds present in the sewage sludge [4–6].

The end-products of the WO process are mainly $CO_2$ and $H_2O$, but the generation of short-chain organic acids, such as volatile fatty acids (VFA), as reaction intermediate compounds is common [7,8]. All of them are easily biodegradable but refractory to oxidation, with acetic acid being the most important among them. Additionally, liquid effluent is free of pathogens, while in the case of nutrients, nitrogen in the form of ammonium and phosphorus in soluble form, as orthophosphates, were usually detected. The solid fraction is stable and mostly inorganic, with heavy metals and insoluble phosphorus content, while the gaseous effluent is free of toxic compounds usually generated in the incineration processes [9]. For all these reasons, wet oxidation treatment facilities have been installed in about 200 industrial-scale plants worldwide to treat municipal and industrial sewage sludge [10–12]. Due to its exothermic nature, which is self-sustained from a certain organic matter content in the sewage sludge, energy recovery is one of the main technology characteristics [13,14].

On the other hand, nutrients play an essential role in WWTPs from both a sustainability and legislative point of view. To avoid eutrophication and pollution in the environment caused by excessive nutrient content in the aqueous medium, the legislation on nutrient concentration limits in WWTPs effluents is becoming more and more restrictive, especially concerning phosphorus discharge.

Phosphorus is widely used as an agricultural fertilizer and is a non-renewable resource obtained from phosphate rock mines. Multiple authors and agencies confirm its depletion in the short–medium term (30–200 years) [15–17]. Furthermore, the high geographic concentration of phosphate rock reserves (up to 72%) in Morocco and Western Sahara is a challenge in terms of environmental issues but also from a political point of view [18–20]. All this, together with the different perspectives of phosphorus consumption linked to more or less conservative population growth projections, allows the existence of multiple studies that predict when phosphate rock reserves will be exhausted, with variations observed from the next 30 to more than 200 years.

All these aspects have driven the need for phosphorus recovery, making sewage sludge a fascinating source for this purpose. After the finding of struvite deposits formed by spontaneous precipitation in the pipes of several WWTPs, this compound began to receive significant interest. Struvite is a white mineral with a crystalline and pyramidal orthorhombic structure. Additionally, it is generally easy to recover and valuable as a slow-release fertilizer [21–23]. The current methods of recovering phosphorus in the form of struvite are mainly carried out with the liquid effluent from the anaerobic sludge digestion since this fraction concentrates large amounts of P-$PO_4^{3-}$ and N-$NH_4^+$ generated by the microorganisms involved in the biological process.

The usual treatments in WWTPs are designed to remove phosphorus from the water line to ensure discharges with a phosphorus concentration below the legislative limits. Usually, the addition of metallic coagulants such as $FeCl_3$, $Ca(OH)_2$, or $Al_2(SO_4)_3$ in the primary treatment and settling after the secondary treatment lead to more than 90% of the phosphorus concentration at the inlet of WWTPs ending up in the sludge line [24–26].

In this way, several industrial-scale applications have been developed to recover phosphorus in the form of struvite using sewage sludge as a precursor. Thus, the AirPrex® process uses the effluent after the anaerobic digestion of the sludge without being dewatered. In contrast, the PHOSHPAQ process treats the liquid fraction of the dewatered sludge. Both treatments require the addition of Mg and, through aeration, achieve $CO_2$ stripping to reach a slightly basic pH (8–8.5), which causes the spontaneous precipitation of struvite in the presence of ammonium [27–29]. At the laboratory scale, the phosphorus recovery as struvite from a liquid effluent after WO treatment of municipal sewage sludge has been studied in several works [30,31] through processes similar to those applied in this work.

Generally, struvite is obtained following Equation (1), using equimolar relations between the three main components ($Mg^{2+}$, $NH_4^+$, $PO_4^{-3}$) in aqueous media. The presence of other compounds, mainly Ca, could compete in the phosphate precipitation reaction at pH values close to neutrality, forming different compounds, such as hydroxyapatite or monocalcium phosphate [32–35].

$$Mg^{2+} + NH_4^+ + PO_4^{-3} + 6H_2O \rightarrow NH_4MgPO_4 \bullet 6H_2O \tag{1}$$

It has been established that struvite precipitation occurs spontaneously at basic pH. Stirring speed, contact time, and temperature are parameters that strongly affect the extraction process. Furthermore, to consider the production of struvite on an industrial scale, other specific parameters must be considered to obtain specific particle sizes [21,23].

Furthermore, after WO treatment, the stabilized and mostly inorganic solid fraction usually contains amounts of inorganic phosphorus that can be recovered. Direct agricultural applications are not possible due to the toxicity attributed to the high presence of heavy metals in the solid fraction. Therefore, several studies have been focused on extracting the phosphorus from the solid matrix into an aqueous one as soluble orthophosphates, similar

to the recovery of incinerated sewage sludge ash (ISSA) and comparable in composition to the solid after the WO process [36–38].

Furthermore, promising results regarding the chemical extraction of phosphorus using the ISSA process after WO have been reported in the literature [39–41]. In this case, phosphorus leaching was studied in an initial approximation to determine the P-extractable from the solid fraction, aiming to maximize its recovery.

Thus, this work aims to evaluate a comprehensive phosphorus recovery process from the liquid and solid fractions generated after a wet oxidation treatment of sewage sludge. Liquid effluent containing dissolved orthophosphates was used to accomplish the struvite precipitation at optimal conditions. Additionally, the solid effluent was treated with acid solutions to maximize the phosphorus leaching.

## 2. Materials and Methods

### 2.1. Wet Oxidation Process

The WO process was carried out in a pilot plant owned by ECOLOTUM, Energía Recuperable S.L., and operating in a Spanish wastewater treatment plant (WWTP).

The sewage sludge was obtained by collecting and mixing the sludge generated after the primary and secondary treatments of the WWTP, as mentioned above. The experiments were accomplished over two weeks, taking different samples of the initial sludge and the final effluent within the reaction time (24 h). As the conditions were kept constant (50 L/h, 350 °C, and 240 bar), the taken samples were mixed to obtain only one input and one output sample, which were characterized and used for the different precipitation experiments.

The reactor operates at high pressure and temperature (240 bar and 350 °C) that were close to and considered as subcritical water conditions at continuous mode, with a volumetric flowrate of 50 L/h and using pure oxygen in excess respect to the stoichiometric amount.

The effluent obtained after the WO process, a mixture of liquid and solid fractions, was collected, conveniently transported, and stored at 5 °C to carry out further experiments. Thus, the solid collected was separated by vacuum filtration on filter paper and was dried in an oven at 105 °C for 12 h before being ground and sieved from 355 to 500 µm.

The inlet sewage sludge and the outlet stream from the WO process were analyzed to evaluate the organic matter and solids content using standard methods [42]. Several parameters have been measured, such as chemical oxygen demand (COD) by spectrophotometry following the colorimetric method using acid-dichromate commercial vials; total solids (TS), total volatile solids (TVS), and total fixed solids (TFS) concentrations were analyzed by gravimetric method. Additionally, the solid after the WO process was studied by X-ray fluorescence (XRF) and micro-elemental analysis to determine heavy metals and other inorganic compounds concentration, as well as the organic compounds percentage, such as C and N content.

### 2.2. Struvite Precipitation Experiments

Struvite precipitation experiments were carried out using Jar Test equipment that allowed us to accomplish several experiments simultaneously. For this purpose, 500 mL of liquid effluent were placed in 600 mL borosilicate beakers, and then, the required amounts of the NaOH, 1 M solution, and the solid $MgCl_2 \cdot 6H_2O$ were dosed. The struvite obtained was separated from the liquid by vacuum filtration on filter paper (0.45 µm), and after washing with ultrapure water, it was dried at room temperature for 24 h before being characterized.

To optimize the operating conditions evaluated, i.e., pH and Mg/P molar ratio, and to maximize the phosphorus recovery in the liquid after precipitation, an experimental design was carried out. The operational requirements selected ranged between pH = 7.5 to 9.5 ± 0.2 and 1 to 4 Mg/P molar ratio, in agreement with previous research studies [21,30].

The orthophosphate content was analyzed in the liquid fraction, both before and after precipitation. Thus, total phosphorous content was determined to verify that the presence

of orthophosphates is the primary form of soluble phosphorus. Both measurements were accomplished by UV-Vis spectrophotometry in a PF11 Macherey-Nagel spectrophotometer using the molybdate/ascorbic acid blue method. Additionally, the pH was monitored using a micropH meter 2002 (Crison Instruments, Barcelona, Spain).

The chemicals used were NaOH (quality analysis, provided by Merck kGaA, Darmstadt, Germany) and $MgCl_2 \cdot 6H_2O$ (99% purity, provided by Panreac).

Overall, the solid obtained was characterized by micro-elemental analysis and inductively coupled plasma optical emission spectrometry (ICP-OES) after acid digestion in 1% $v/v$ of nitric acid and heating up to 100 °C. In addition, some XRF analyses were performed to discard the co-precipitation of other ions in the solid.

### 2.3. Solid Leaching Experiments

In this research, the solid fraction was dried, grounded, and sieved. For each experiment, the solid fraction and ultrapure water were placed in a 250 mL Erlenmeyer flask at constant magnetic stirring (450 rpm) and room temperature conditions, measuring pH in continuous mode.

To characterize the solid after the WO process and determine the fractions of phosphorus bound to other compounds, a sequential extraction process was carried out using 1 g of each solid. Between each extraction step, the solid was washed with 20 mL KCl, 1 M solution and filtered again. This procedure was accomplished following the next steps [41,42]:

- $NaHCO_3$ fraction. A total of 20 mL of $NaHCO_3$, 0.5 M to determine the P fraction weakly bound.
- NaOH fraction. A total of 20 mL of NaOH, 0.1 M to determine Al and Fe bound P.
- HCl diluted fraction. A total of 20 mL of HCl, 1.0 M to define Ca bound P.
- HCl concentrated and hot fraction. A total of 12 mL of HCl, 10 M in a water bath at 80 °C to extract stable P. This fraction needs a large amount of energy to be recovered.

Preliminary tests were accomplished using an HCl solution to evaluate P leaching by varying contact time, acid concentration, and liquid/solid ratio, using 1 g of solid. After that, an experimental design was performed considering four different acids at different concentrations to determine the optimal parameters of phosphorus extraction. In addition, some leachates were also analyzed to identify the presence of heavy metals or toxic compounds.

The chemicals used were KCl (chemically pure crystal, provided by Probys), $NaHCO_3$ (chemically pure, provided by Solvay), HCl (fuming $\geq$37% quality analysis, supplied by Honeywell Fluka), $HNO_3$ (purity 65–67%, provided by Sigma Aldrich), and citric acid (solid purity >99%, obtained by ProBys).

The solid used as raw material was characterized by X-ray fluorescence (XRF) and micro-elemental analysis techniques, while total phosphorus content was measured after digestion in aqua regia (1:3 $HNO_3$:HCl $v/v$), using the ascorbic acid blue method [42]. The phosphorus content as orthophosphate was measured in the liquid before and after the leaching experiments. In some cases, total phosphorus after leaching was analyzed to compare how much phosphorus was extracted in the orthophosphate form.

## 3. Results and Discussion

### 3.1. Characterisation of Sewage Sludge

Several experiments by wet oxidation (WO) process using sewage sludge from a wastewater treatment plant (WWTP) located in Spain were carried out. The characterization results of both the initial sludge and the effluent after the oxidation process are shown in Table 1.

**Table 1.** Characterization of inlet sewage sludge and mixed outlet effluent of the oxidation process.

| Parameters | Inlet Sludge | Effluent Outlet | Conversion (%) |
|---|---|---|---|
| COD (g $O_2$/L) | $17.90 \pm 3.5$ | $3.11 \pm 0.05$ | 82.71 |
| TS (g/kg) | $27.80 \pm 2.4$ | $7.05 \pm 1.5$ | 74.64 |
| TVS (g/kg) | $12.10 \pm 2.4$ | $0.60 \pm 0.1$ | 95.04 |
| TFS (g/kg) | $9.20 \pm 2.4$ | $6.70 \pm 1.5$ | 27.17 |

Subsequently, a mixed effluent was separated into liquid and solid fractions used as raw materials for both extraction processes. The characterization parameters of both the liquid effluent and solid fraction have been collected in Table 2.

**Table 2.** Characterization parameters of the effluent liquid and solid fraction.

| Parameters | Liquid Effluent |
|---|---|
| COD (g $O_2$/L) | $3.1 \pm 0.05$ |
| Total, P (mg P/L) | $93.1 \pm 4$ |
| P-$PO_4^{3-}$ (mg P/L) | $86.1 \pm 5$ |
| Total, N (mg N/L) | $975 \pm 22$ |
| N-$NH_4^+$ (mg N/L) | $848 \pm 17$ |
| $Mg^+$ (mg Mg/L) | $2.5 \pm 1$ |
| $Ca^{2+}$ (mg Ca/L) | $7.1 \pm 1$ |
| Total, P (mg P $g_{solid}^{-1}$) | 68.20% (in solid effluent) |
| **Elemental analysis** | **Compounds (%)** |
| C | $0.88 \pm 1$ |
| H | $0.92 \pm 1$ |
| N | $0.69 \pm 1$ |
| **XRF analysis** | **Oxides (%) [a]** |
| Si | 14.01 |
| Fe | 12.79 |
| Al | 7.39 |
| Ca | 5.84 |
| Mg | 2.40 |
| K | 1.80 |
| $\sum$ (Ti, Na, Zn, Cu, Mn, Ga, Pb, Ba, Sr, Co, Cr, Zr, Ni, Rb, Sn) | 1.39 |
| CL | 51.89 |

Note: [a] R.M.S. (Root Mean Square) = 0.01. CL: Calcination loss at 1000 °C.

As can be observed, in the wet oxidation sewage sludge treatment, conversion values of up to 83% in chemical oxygen demand (COD) and up to 75% in total solids (TS), of which up to 95% were total volatile solids (TVS) associated with organic matter content, were achieved. Thus, the fixed total solids (TFS) are related to the inorganic compounds' presence, making phosphorus the most valuable.

The organic matter content in the liquid effluent after the oxidation process is mainly composed of short-chain fatty acids, with acetic acid being the most representative compound. Therefore, the liquid effluent containing highly soluble and biodegradable organic matter could guarantee a possible use in a subsequent biological process.

Furthermore, the solid effluent is mainly inorganic, with a total carbon content lower than 1%. The main compounds of the solid fraction were micronutrients, such as Si, Fe, and Al oxides, but a high phosphorus content was also found.

This phosphorus content, both in the liquid and solid effluents, motivated the extraction studies as an integral mechanism to improve the oxidation process from a sustainable point of view. Thus, WO can be understood as a process of sludge pretreatment that makes subsequent phosphorus recovery easier, in this case, by transforming phosphorus into soluble species in an aqueous medium that could be relatively quickly recovered.

The calcination loss at 1000 °C was 51.89% and includes humidity loss, organic matter, dehydroxylation of phyllosilicates, decarbonation of carbonates, etc.

### 3.2. Struvite Precipitation Experiments

Struvite chemical precipitation was the operation selected to recover phosphorus as orthophosphates in an aqueous medium. For this purpose, several experiments were performed by varying the two main operating parameters, i.e., pH and the Mg/P molar ratio. The selected operating conditions are shown in Figure 1.

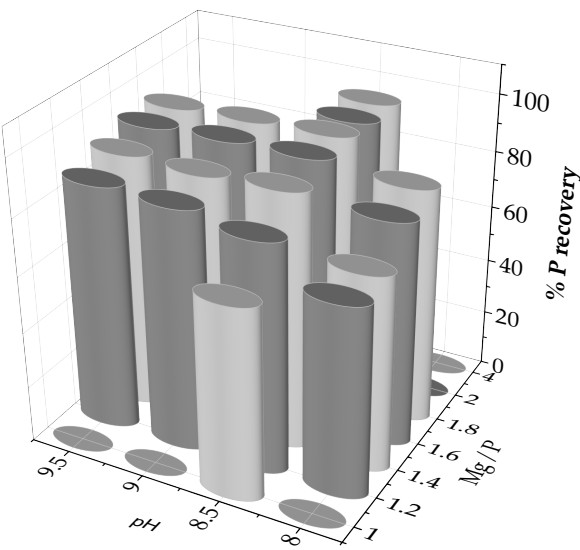

**Figure 1.** Phosphorus recovery results by varying pH and Mg/P molar ratio.

From these experiments, it could be stated that pH is the limiting parameter of the process, considering that at pH 7.5 and lower values, no precipitated phosphorus was obtained. In addition, phosphorus recovery increased up to the pH value of 8.5, while when this value was exceeded, P recovery did not increase and even slightly decreased. Therefore, pH 8.5 has been established as the optimum value for P recovery.

On the other hand, the Mg/P molar ratio behaves more linearly, always favoring P recovery when the $Mg^{2+}$ excess increases, establishing a direct relationship between both parameters. Therefore, the P recovery percentage was established as a function of the Mg/P molar ratio at a constant pH of 8.5, as can be seen in Figure 2a. Thus, the phosphorus recovery reached a maximum when the Mg/P molar ratio was 1.8 (%P recovery slightly higher than 95%); thus, this value increased from an Mg/P ratio of 1.0 and then slightly decreased at higher ratio values (Mg/P = 2.0, and 4.0). Consequently, the variation of pH in the experiments was carried out at an Mg/P molar ratio of 1.8 (Figure 2b).

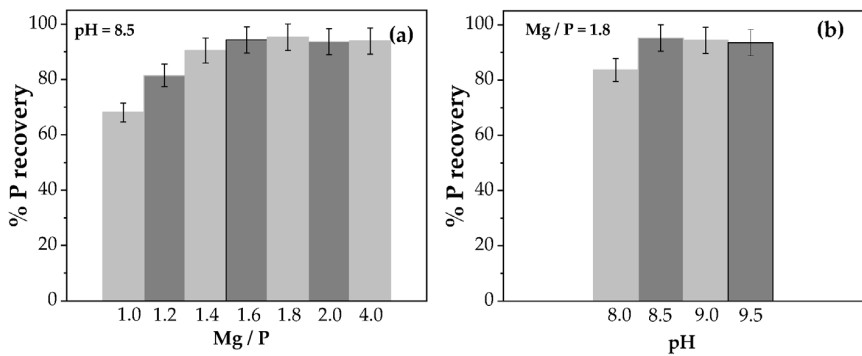

**Figure 2.** Phosphorus recovery at various (**a**) Mg/P ratios for constant pH and (**b**) pHs for constant Mg/P ratio.

From this moment on, the struvite obtained under the operating conditions of pH 8.5 and a Mg/P molar ratio of 1.8 will be referred to as optimal struvite. Still, it should be highlighted that it has been defined only in the operation conditions range studied and taking into account only the phosphorus recovery. An additional economical study will be necessary to determine the parameters that, in addition to maximizing the P recovery, could minimize the costs of the process [43].

After the precipitation and filtration stages, the obtained solid was characterized. The three main components' (P, N, and Mg) concentration values were determined by the ICP-OES technique and compared to the literature reference values that define the struvite-precipitated purity. At the same time, the elemental analysis technique has been used to determine the organic matter content of the solid. In addition, an ICP-OES analysis of the optimum struvite has been carried out, including the presence of other co-precipitated compounds. These results are shown in Table 3.

**Table 3.** Purity, organic matter content, and ICP-OES analysis of optimal struvite.

| pH | Molar Ratio Mg/P | %P Purity | %N Purity | % Mg Purity | % C | Compound | Concentration ($\mu g/g_{struvite}$) |
|---|---|---|---|---|---|---|---|
| 8.0 | 1.2 | 92.7 | 91.1 | 65.7 | 0.12 | Si | 19,255 ± 1759 |
| 8.0 | 1.4 | 92.7 | 91.1 | 72.7 | 0.17 | Al | 1687 ± 185 |
| 8.0 | 1.6 | 91.9 | 91.2 | 73.7 | 0.14 | Fe | 563.5 ± 53 |
| 8.0 | 1.8 | 92.7 | 91.2 | 73.7 | 0.10 | As | 123.5 ± 12 |
| 8.5 | 1.2 | 91.9 | 90.0 | 73.7 | 0.17 | S | 72.0 ± 7 |
| 8.5 | 1.4 | 91.9 | 89.8 | 73.7 | 0.18 | Mn | 46.5 ± 5 |
| 8.5 | 1.6 | 91.9 | 89.7 | 72.7 | 0.17 | Zn | 29.0 ± 3 |
| 8.5 | 1.8 | 92.3 | 90.0 | 72.7 | 0.22 | Cu | 13.6 ± 1.4 |
| 8.5 | 2.0 | 91.1 | 88.4 | 72.7 | 0.15 | Cr | 12.2 ± 1.2 |
| 8.5 | 4.0 | 79.2 | 76.7 | 70.5 | - | Cd | <10 Below the limit of detection (LOD). |
| 9.0 | 1.2 | 90.3 | 89.7 | 72.7 | 0.20 | Mo | |
| 9.0 | 1.4 | 89.5 | 89.1 | 72.7 | 0.20 | Ni | |
| 9.0 | 1.6 | 88.0 | 87.6 | 71.7 | 0.26 | Pb | |
| 9.0 | 1.8 | 87.2 | 87.0 | 69.7 | 0.28 | Hg | |
| 9.5 | 1.2 | 88.0 | 89.3 | 68.7 | 0.33 | | |
| 9.5 | 1.4 | 87.2 | 87.4 | 68.7 | 0.33 | | |
| 9.5 | 1.6 | 88.0 | 86.9 | 70.7 | 0.35 | | |
| 9.5 | 1.8 | 88.0 | 87.6 | 66.7 | 0.35 | | |

As the pH increased, the concentration of both phosphorus and nitrogen decreased in the struvite, slightly reducing the purity values of the precipitate. The reduction could be considered almost negligible, only 5% in both compounds, and the experiments at pH 8.0 provided the precipitate with the highest content in P and N. Thus, the purity values of both compounds were very high, greater than 90% under optimum conditions. The magnesium content followed a similar trend but with a much lower richness than pure struvite, reaching up to 74%.

In the range of the Mg/P molar ratio from 1.2–1.8, the composition of any of the three compounds (P, N, Mg) in the precipitate did not follow any trend. Similar contents were obtained in all the experiments carried out. In the experiments at pH 8.5, where a more significant excess of magnesium was used, it was observed that the contents of the three compounds that formed part of the precipitate were reduced as the Mg/P molar ratio increased, becoming more pronounced at a molar ratio of 4.

On the other hand, the organic matter content behaved oppositely. It was minimal at pH 8 and increased when the pH was raised. However, the percentage of carbon in struvite did not exceed 0.35%. This aspect is essential for the possible application of struvite as a fertilizer.

Regarding the composition measured in the optimal struvite, which can be considered impurities, the sum of all of them resulted in 2.2%, with Si being the main compound, reaching 1.93%. In addition, micronutrients such as Fe, Cu, and Zn were detected. However, small amounts of toxic or dangerous agricultural compounds, such as Cr, As, or Hg, were also found [30,44].

According to recent European legislation, struvite is no longer considered waste but a potential fertilizer product [45], offering a new alternative to obtain phosphorus in this way.

### 3.3. Preliminary Experiments

The solid fraction characterization results are shown in Table 2, and the preliminary solid leaching tests were performed following the sequential extraction process described above. The results of these experiments, conducted in duplicate, are presented in Table 4.

**Table 4.** Chemical phosphorus extraction. Sequential extraction in percentage.

| Steps | Struvite Extraction (%) |
|---|---|
| Not extracted | 5 |
| NaHCO$_3$ fraction | 1 |
| NaOH fraction | 11 |
| HCl diluted fraction | 53 |
| HCl concentrated and hot fraction | 30 |

As can be seen in Table 4, most phosphorus in the solid (53%) corresponded to that extracted with dilute acid (HCl solution), related to Ca-bound phosphorus. Then, 30% was stable phosphorus extracted with concentrated HCl solution at hot conditions. Thus, more than 80% of the phosphorus was extracted using acid solutions, while the bases account for only 12%, of which 11% corresponded to the strong base NaOH, and 1% corresponded to the weak base NaHCO$_3$ solution. Only a 5% fraction could not be extracted.

Considering these preliminary results, leaching tests will be carried out using dilute HCl solutions. Some operating parameters, such as the contact time between phases, the acid concentration, and the liquid/solid ratio used, were modified to determine their effect on phosphorus leaching. In addition, both total phosphorus and phosphorus in the form of orthophosphates will be analyzed in the liquid effluent to determine how much was extracted in this form.

The leaching test results with varying contact times, comparing total phosphorus and orthophosphate (P-PO$_4^{3-}$), are shown in Figure 3. The influence of acid concentration and liquid/solid ratio on phosphorus recovery by leaching can be seen in Figure 4. The main factor defining the efficiency of each experiment is the amount of phosphorus leached from the solid to the liquid phase.

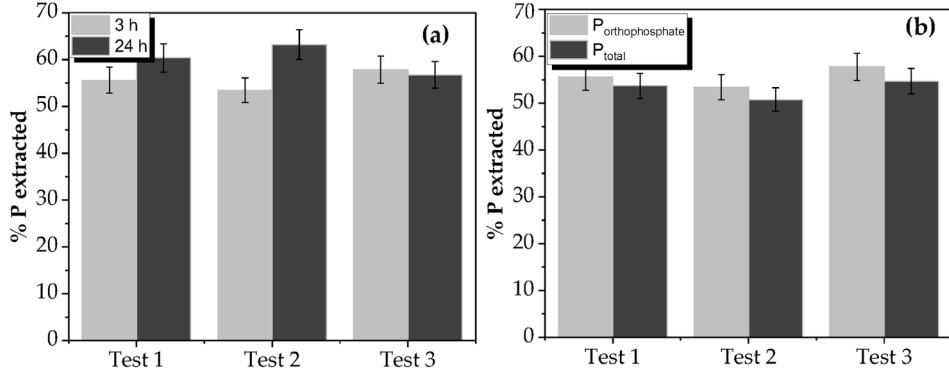

**Figure 3.** Phosphorus extracted varying (**a**) contact time; (**b**) comparing phosphorus measurement.

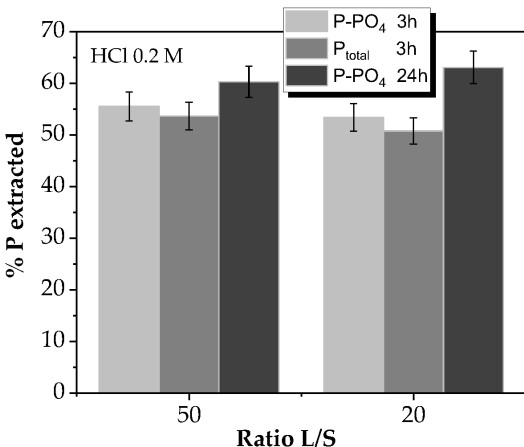

**Figure 4.** Phosphorus extracted varying liquid/solid ratios.

Increasing the contact time between the two phases from 3 to 24 h did not significantly increase the phosphorus leaching, although, in one of the tests, it reached up to 10% (Figure 3a). Despite this, increasing the contact time did not imply a considerable increase in the leached phosphorus quantity. So, it could be assumed that the increase in the operation time up to 24 h was not necessary since the difference in the %P extracted could be considered negligible in most cases.

As can be seen in Figure 3b, orthophosphates are the main form in which phosphorus has been obtained after the leaching process of the solid. This fact is important since this leached phosphorus could be recovered as struvite. Even measurements have pointed out a higher presence of orthophosphates than total phosphorus (Figure 3b), which can be assumed as a standard deviation of the measurements.

Moreover, as can be observed in Figure 4, decreasing the liquid-to-solid ratio (L/S) from 50 to 20 only reduced the phosphorus leaching by over 2% on average, so it could be considered that the amount of liquid is, in both cases, in enough excess, not being a limiting factor in the process. So, fortunately, the phosphorus extraction process has not been affected by the ratio liquid/solid value at the range studied.

It was found that the maximum extracted phosphorus value using HCl 0.2 M solution was 63%, with a contact time of 24 h. Reducing this parameter to 3 h, the maximum P recovery was 60%, at the same operational conditions (Figure 4). However, the acid dosage is the most critical parameter in the phosphorus extraction process, both from a technical and economic point of view. Hence, its reduction is essential for the efficiency of the process.

The objective of studying the influence of the different HCl solution concentrations on phosphorus recovery is to evaluate the lowest amount of acid that could be used in an efficient way in the process. As mentioned before, phosphorus leaching tests using 0.5 and 0.2 M HCl solutions led to similar values, with deviations of 2%, which can be considered negligible. This indicated that the minimum acid dose to maximize the phosphorus leaching recovery might be even lower than 0.2 M HCl, so, in this regard, experiments with concentrations lower than 0.2 M were conducted.

To discuss the decreasing of acid solution concentration for an efficient phosphorus extraction process, several experiments were carried out using HCl solutions at concentrations considerably lower than those already studied, as shown in Figure 5. It can be seen that by reducing the acid concentration by almost 10 times (0.0234 M), a relatively high phosphorus recovery was obtained (45%).

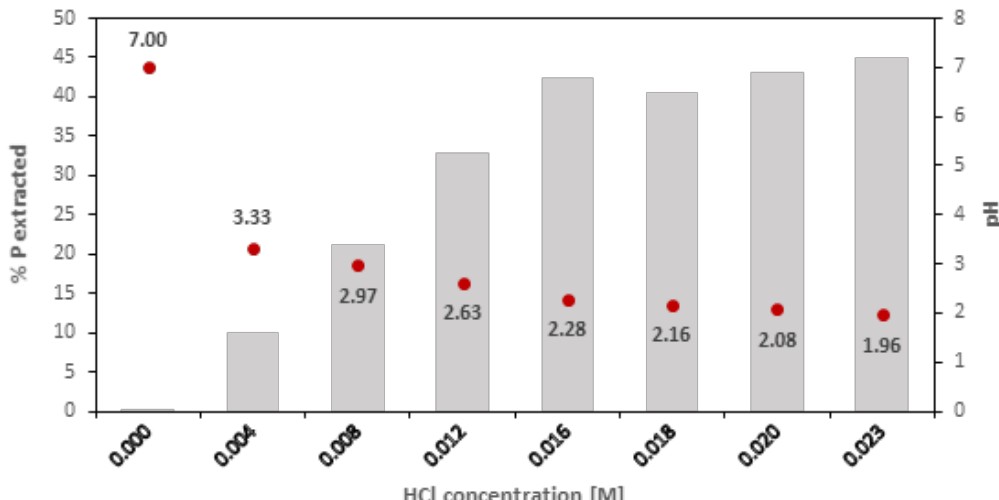

**Figure 5.** Influence of lower values of HCl solution concentration and pH on phosphorus extracted.

In this way, the ratio of mL HCl/gP extracted was evaluated to compare the percentage of phosphorus extracted as a function of the amount of acid solution used. In a quick mass balance, the phosphorus recovery using HCl 0.0234 M solution needed 3.13 mL HCl/gP$_{recovered}$, while for an HCl 0.2 M solution, a higher quantity of HCl solution would be necessary (19.28 mL HCl/gP$_{recovered}$).

An economic study not included in this work is necessary to determine which option involved a more cost-effective process. However, based on these data, decreasing the acid dose by 10 times while losing 15% recovery will be more economically profitable for the process. Therefore, the experimental design has been focused on analyzing phosphorus recovery at these low HCl solution doses using the above-mentioned ratio and molar concentrations of different acids for comparison.

### 3.4. Solid Phosphorous Extraction

Taking into account the previous results, the phosphorous extraction was carried out by comparing four different acids (hydrochloric, nitric, sulfuric, and citric acid) in varying concentrations (0.00794 to 0.0234 M) and monitoring pH, while the contact time and liquid/solid ratio were kept constant (24 h, and L/S ratio of 50). The obtained results are shown in Figure 6.

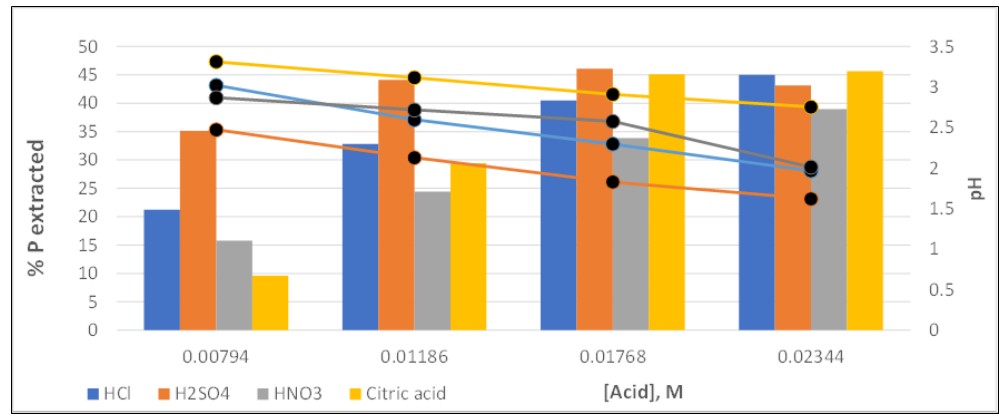

**Figure 6.** Comparison of P extracted at experimental design conditions, monitoring pH.

As can be seen in the figure, sulfuric acid achieved the highest phosphorus recovery values, with more significant differences from the other acids at low concentrations. This is because this acid reduces the pH solution the fastest.

On the other hand, nitric acid provided the lowest recovery values out of all the concentrations studied. Concerning hydrochloric and citric acid, the first achieved higher recoveries than the second at low concentrations, while the trend seemed to be reversed in the higher acid concentrations range (specifically, for 0.01768 M, Figure 6). Citric acid led to the highest phosphorus recovery value (45.66%) at the highest solution concentration (0.0234 M), although the differences found were small, less than 3%, among the other three acids tested (hydrochloric, sulfuric, and nitric) at the same acid concentration.

As the concentration of each acid increased, in the case of sulfuric, hydrochloric, and nitric acids, the ratio of acid used/phosphorus recovered slowly increased. This indicated that as phosphorus was extracted, more and more acid was needed, making the extraction process less efficient from the point of view of the acid dosage used.

Moreover, the difference between them decreased as the used acid dose increased, reaching similar final values among the four acids (~0.7–0.8 mol acid/g $P_{recovered}$). This fact indicated that the phosphorus recovery is not influenced by the type of acid when high acid doses are being used.

So, at low doses, sulfuric acid is the most profitable in terms of the ratio of acid used and the amount of phosphorus recovered. In contrast, as the dose increased, the obtained results were very similar.

To determine the most profitable dose in a potential industrial scale-up, an economic balance is required, and, in addition to the acid concentration, its cost must be evaluated. This aspect will be considered further in future works.

## 4. Conclusions

Regarding the struvite precipitation, in the liquid effluent, 86.11 mg P/L was found to be orthophosphates that were easily recoverable by struvite precipitation. At the same time, in the solid fraction, 68.2 mg $P/g_{solid}$ was measured, which could be also extracted as orthophosphates by leaching using an acid solution. The operating parameter values that maximized the phosphorus recovery (95.24%) were the molar ratio Mg/P = 1.8 and pH = 8.5. The purity of the obtained precipitates was higher (P, N, and Mg content) when the precipitation experiments were carried out at low pH (8 and 8.5), decreasing these contents with increasing pH. Therefore, no similar trend was observed for the Mg/P ratio.

On the other hand, in the phosphorus extraction from the solid, the pH showed a clear correlation with the phosphorus extracted from the solid. The highest extraction percentages were obtained at the lowest pH values tested, independently of the acid employed. Increasing the leaching contact time from 3 to 24 h did not significantly increase the phosphorus recovery. Finally, among the acids studied, sulfuric acid achieved the highest phosphorus recovery values at the lowest acid doses, showing significant differences in comparison to the other acids.

Future research should be conducted to optimize the operating parameters while applying economic criteria.

**Author Contributions:** Conceptualization, J.G., S.Á.-T., J.C. and B.H.; methodology, J.C. and S.Á.-T.; software, J.C.; validation, J.C.; formal analysis, J.G., S.Á.-T., J.C. and B.H.; investigation, J.C.; writing—original draft preparation, J.C., S.Á.-T. and J.C.; writing—review and editing, J.G., S.Á.-T., J.C. and B.H.; visualization, J.G., S.Á.-T., J.C. and B.H.; supervision, J.G., S.Á.-T. and B.H.; project administration, J.G. and B.H.; funding acquisition, J.G. and B.H. All authors have read and agreed to the published version of the manuscript.

**Funding:** This research received funding from MICINN, through the CATAD3.0 project PID2020-116478RB-I00. In addition, the authors acknowledge funding from the Comunidad de Madrid (Spain) through the Industrial Ph.D. projects (IND2017/AMB-7720 and IND2019/AMB-17114), as well as the REMTAVARES Network (S2018/EMT-4341), the European Social Fund and the Spanish Center for Technological and Industrial Development (CDTI).

**Data Availability Statement:** Not applicable.

**Acknowledgments:** This work was supported by MICINN, through the CATAD3.0 project PID2020-116478RB-I00. In addition, the authors acknowledge funding by the Comunidad de Madrid (Spain) through the Industrial Ph.D. projects (IND2017/AMB-7720 and IND2019/AMB-17114), as well as the REMTAVARES Network (S2018/EMT-4341), the European Social Fund and the Spanish Center for Technological and Industrial Development (CDTI). J.C.J. is grateful to the CAM for granting his Industrial Doctorate.

**Conflicts of Interest:** The authors declare no conflict of interest.

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
