# Peer review of "Phosphorus Recovery from Sewage Sludge as Struvite"

_water, doi:10.3390/w15132382_

Round 1

Reviewer 1 Report

The manuscript “Circular Economy Model through Phosphorus Recovery to Obtain Struvite from Sewage Sludge” reports the results of experiments carried out to recover P from the liquid and solid fractions of the effluent from wet oxidation of sewage sludge. Struvite precipitation and P-leaching were studied, respectively. The topic is of clear interest for the readers of Water but several major issues emerged during the revision process and I have to suggest Rejection for the manuscript. Please find below the major comments, as well as other minor points.

Major comments:

(1) English quality is poor throughout the manuscript. There are lot of typos (i.e., ° should be used before Celsius), complex syntax sometimes is making the reading and comprehension hard, grammar mistakes (i.e., “before BE CHARACTERIZED”). In addition, the presentation quality is poor due to the continuous change of the tenses (from past to present to future!) and due to other minor issues (i.e., use of “our”).

(2) The manuscript is completely missing even a simple statistical analysis. How many samples were analysed? Were the experiments replicated? Were the analyses replicated? Which are the standard deviations? Do you carry out at least a regression analysis when you say “linearly” (Line 241)? Scientifically, the experiment is not sounding.

(3) There are some experiments that were not described in the Materials and Methods section, and they were neither mentioned in the aim of the research. I refer to sections 3.3 (Lines 344-3358) and Section 3.4. Which is the aim of Section 3.4? Those results, and the way they were obtained (?) and presented, are not scientifically sounding. Conclusions from this part are not supported by the results.

Minor comments:

- Title: remove the final “.”

- Introduction is too long and should be shortened. I.e., Lines 52-54 should be cut, and Lines 69-99 have to be summarised.

- Lines 198-199 are not clear and should be rephrased.

- Table 3: check unit of Total P cause the “%” close to Solid Effluent is confusing.

- Lines 226-233 can be cut.

- Figure 2 is repetitive of figure 1, please cut.

- Table 4 is confusing (the last two columns) and should be modified to make it clearer.

- Lines 289-291 can be cut.

- Figure 3 is useless, please cut.

- Conclusions are just a long repetition of Results and need to be completely edited.

English quality is poor throughout the manuscript. There are lot of typos (i.e., ° should be used before Celsius), complex syntax sometimes is making the reading and comprehension hard, grammar mistakes (i.e., “before BE CHARACTERIZED”). In addition, the presentation quality is poor due to the continuous change of the tenses (from past to present to future!) and due to other minor issues (i.e., use of “our”).

Reviewer 2 Report

This paper deals with the evaluation of the phosphorus recovery that can be obtained from the liquid and solid fractions resulting from the wet oxidation treatment of sewage sludge. Liquid effluents containing dissolved orthophosphates (Mg/P molar ratio, pH, struvite purity, etc.) were used to obtain optimum conditions for struvite precipitation. The solid waste was also treated with acid solutions to maximise phosphorus leaching. All experiments were conducted under laboratory conditions.

Suggestions:

1) I am not sure if the title of the paper corresponds to the content of the paper. In the case of the circular economy, in addition to the economic evaluation, a technological scheme should be presented, with the flows of inputs and outputs during the processing of liquid and solid streams.

2) Chemical formula of the struvite formaiton – in Equation (1) should be written by the following chemical formula: Mg2+ + NH4+ + PO4-3 + 6H2O → NH4MgPO4•6H2O

Otherwise, the article presents the results of its work nicely. Maybe it could have been supplemented with a few photos.

Reviewer 3 Report

The title of the manuscript does not match with research/experimental results of this manuscript. Circular economy model related research result is not present in this manuscript. It is only phosphorus recovery from sewage sludge and it does not have any specific novelty.

If authors go for further correction, then following suggestions are suggested to follow.

1. A suggested title can be similar to "Phosphorus recovery from sewage sludge as struvite.

2. Line 40: reference is needed.

3. Line 59: reason is required for any limitation.

4. from the begining to the end: circular economy and valorization wards are suggested to remove.

5. Line 95: "they are kept constant" - need constant value.

6. Line 111: rewrite it.

7. Line 123: Put the value of reaction time and constant condition.

8.  Line 126: Mention the high temperature and pressure value.

9. Line 196: change the subsection 3.1 and rename it to characterization of sewage sludge.

10. Line 211: Change the tittle of Table 2 and Table 3 and combine them into one (elemental analysis of ------- is sughgested).

11. Figure 1: correct the legend (mg/P to Mg/P).

12. Figure 2: title can be changed to "Phosphorus recovery at various a) Mg/P ratio for constant pH and b) various pH for constant Mg/P ratio.

13. Table 4 (will be Table 3) suggesting to rearrange. As P is shown in Figure 2, N, Mg, C and other elements can represent in concentration (dry) basis (not in %). Table title should be short and rename it.

14. Section 3.3: Figure 3 is not understandable. Table form may be better or change the figure 3 into an understandable way.

15. Figure 4 (b) is not clear.

16. Change the title of Figure 5 for clear understanding [(a) can keep and (b) can go to figure 6].

17. Figure 5 (b) and figure 6 can be combined into one figure with same way of  orientation.

18. Section 3.4 can be renamed. Figure 8 can be removed and rewrite the section accordingly.

19. Rewrite the conclusion in short and specific way.

The manuscript can send to a native English language editor for English check up.

Round 2

Reviewer 1 Report

The authors have modified the manuscript accordingly to most of the reviewer comments. Anyway, I still recognize that statistics are missing from the tables and some graphs. Furthermore, Conclusion should be rewritten avoiding the use of numbered list.

English is fine, minor editing needed.

Reviewer 3 Report

Check Table 2  second and third portion (chemical analysis) and XRF analysis. It should be clear to general people (all together shoul be 100%).

Clean figure 4 and Figure 5.

Overall english check by a native english language writer will be good for the manuscript.
